# The High Pressure Preservation of Honey: A Comparative Study on Quality Changes during Storage

**DOI:** 10.3390/foods13070989

**Published:** 2024-03-24

**Authors:** Hana Scepankova, Juraj Majtan, Leticia M. Estevinho, Jorge A. Saraiva

**Affiliations:** 1LAQV-REQUIMTE, Department of Chemistry, Campus Universitario de Santiago, University of Aveiro, 3810-193 Aveiro, Portugal; hana.scepankova@ua.pt (H.S.); jorgesaraiva@ua.pt (J.A.S.); 2CIMO, Mountain Research Center, Polytechnic Institute of Bragança, Campus de Santa Apolonia, 5300-252 Bragança, Portugal; 3Laboratory of Apidology and Apitherapy, Department of Microbial Genetics, Institute of Molecular Biology, Slovak Academy of Sciences, Dubravska Cesta 21, 845 51 Bratislava, Slovakia; juraj.majtan@savba.sk; 4Department of Microbiology, Faculty of Medicine, Slovak Medical University, Limbova 12, 833 03 Bratislava, Slovakia; 5SusTEC, Associate Laboratory for Sustainability and Technology in Mountains Regions, Polytechnic Institute of Bragança, Campus de Santa Apolonia, 5300-253 Bragança, Portugal

**Keywords:** honey, HMF, diastase, antioxidant activity, thermal treatment, pasteurization, microbial quality

## Abstract

In commercially available honey, the application of a heat treatment to prevent spoilage can potentially compromise its beneficial properties and quality, and these effects worsen with extended storage. The high-pressure processing (HPP) of honey is being explored, but its long-term impact on honey quality has not been characterised yet. This study evaluated the effects of HPP and thermal processing on the microbial load, physicochemical quality (i.e., hydroxymethylfurfural content and diastase activity), and antioxidant capacity of honey after treatment and following extended storage (6, 12, and 24 months) at 20 °C. Pasteurization (78 °C/6 min) effectively eliminated the microorganisms in honey but compromised its physicochemical quality and antioxidant activity. HPP initially showed sublethal inactivation, but storage accelerated the decrease in yeasts/moulds and aerobic mesophiles in honey (being <1 log CFU/g after 24 months of storage) compared to unprocessed honey and honey thermally treated under mild conditions (55 °C/15 min). The physicochemical characteristics of the quality of HPP-treated honey and raw unprocessed honey did change after long-term storage (24 months) but remained within regulatory standards. In conclusion, HPP emerged as a more suitable and safe preservation method for *Apis mellifera* honey, with a minimal risk of a loss of antioxidant activity compared to traditional industrial honey pasteurization.

## 1. Introduction

Honey is an attractive functional food possessing health-promoting properties. Marketed honey needs precisely defined composition criteria, according to the European Honey Directive [1] and the Codex Alimentarius Standard for Honey [2]. Commercial honey undergoes heat processing to eliminate microorganisms, such as osmophilic (sugar-tolerant) yeasts, to avoid spoilage [3], but also to minimize the risk of fermentation during storage, as it brings down the honey’s moisture content to a safe limit and delays crystallization. However, honey’s functional activity (e.g., antioxidant and antibacterial activity), organoleptic properties (colour and flavour), and enzymatic activities (e.g., diastase activity) may be diminished by thermal treatment [4,5]. In addition, undesirable compounds are formed due to heating, as is the case with hydroxymethylfurfural (HMF) [6].

From the customers’ point of view, there is a high demand and preference for liquid honey. However, some types of raw honey naturally crystallise faster than others due to their high glucose content and/or pollen grains [7]. The heating of crystallised honey is a common practice among honey producers to achieve a liquid state of honey. On the other hand, there is also increased consumer demand for minimally processed honey or raw honey that possesses higher health-promoting activities in comparison with traditionally processed honey [4].

The deteriorating quality of commercial honey is caused by thermal pasteurization and long-term storage, with this being highly dependent on the severity of the thermal treatment applied in the pasteurization process [4]. Changes in the quality of honey caused by heating/long-term storage can be evaluated by measuring the legislative quality criteria, namely the level of diastase activity and HMF content, which also indicate the freshness of the honey [1,2]. Numerous studies have characterized the effect of thermal treatments under different conditions (temperature and length of processing) on the quality criteria of honey. Although it is evident that the thermal treatment of honey increases the formation of HMF and reduces the enzymatic activity of diastase, in some cases the permissible level of both was not exceeded [8,9,10,11,12,13]. On the other hand, certain biological properties, like antibacterial activity, are more susceptible to degradation during thermal treatment [5,14,15]. Due to the unfavourable side effects of thermal pasteurization, other approaches are being tested to avoid honey spoilage and fermentation while maintaining its health-promoting properties [16,17].

High pressure processing (HPP) has become an increasingly commercially implemented technology for the cold pasteurization of foods worldwide. In HPP, foods are exposed to ultra-high pressure (100–600 MPa) for a few minutes to inactivate the microorganisms present in the food and certain enzymes of interest without destroying the quality components that are normally affected during heat treatment. Numerous studies have evaluated the effect of HPP on foods and on their storage to evaluate the pasteurization effect and possible increases or losses of the quality characteristics of foods after applying HPP, in comparison with unprocessed samples or samples subjected to conventional pasteurization treatments [18,19]. Recently, HPP has become the alternative for thermal food processing since it maintains the nutritional and functional properties of foods while ensuring the microbial quality and safety of the product (reviewed in [20]). There is evidence that HPP may be effective in improving the nutritional value [21], antioxidant activity [22,23], and antibacterial activity of honey [24,25], and maintaining these increased values during storage (up to 12 months). However, most of the conducted studies used manuka honey (*Leptospermum Scoparium*), the composition and properties of which, at least in some important minor constituents, differ from that of non-manuka honey. To date, only a few studies have characterized the effect of HPP on *Apis mellifera* non-manuka honey [16,23,26], and there is only scarce information about the microbial and physicochemical qualities of honey subjected to HPP and stored for a prolonged period of time (1–2 years). Similarly, very few studies have investigated the effect of HPP on stingless bee honeys [27,28].

Therefore, this study aims to characterize the effects of HPP and traditional thermal processing (TP) on multifloral honey from Portuguese mountain pastures. The study evaluate the changes in the (i) microbial contamination (aerobic mesophiles, moulds, and yeasts), (ii) HMF content and enzymatic diastase activity, and, finally, (iii) total polyphenol content (TPC) and antioxidant activity of HPP/TP-treated and untreated honey samples after 6, 12, and 24 months of storage.

## 2. Materials and Methods

### 2.1. Honey Sample

Raw honey from western honeybees (*Apis mellifera*) was kindly donated by a beekeeper from an apiary located in the Trás-os-Montes region of Portugal. The honey was gathered from the wild mountain pastures of northeastern Portugal, half-deserted and with practically no intensive agricultural activity present, with extremely rich spontaneous vegetation. The honey extraction was carried out by centrifuging the honey in a honey extractor, and the maximum temperature used throughout the whole honey procedure was 28 ± 2 °C. All honey samples were taken from the same honey batch.

### 2.2. Determination of the Honey’s Botanical Origin

The honey samples’ pollen was analysed using the methodology described by Erdtman, (1969) [29]. Ten grams of the sample was dissolved in 20 mL of distilled water. The obtained mixture was centrifuged at 2000 rpm for 5 min (Eppendorf Centrifuge 5804R, Hamburg, Germany) and the supernatant was discarded. After that, 5 mL of glacial acetic acid (Merck, Darmstadt, Germany) was added and centrifuged at 2000 rpm for 5 min. Then, acetolysis of the pollen sediments was carried out, using a mixture of 9:1 of acetic anhydride (Panreac Applichem, Barcelona, Spain) and sulphuric acid (José Manuel Gomes dos Santos, Lisbon, Portugal), in a water bath at 100 °C for 2 min. The mixture was centrifuged at 2000 rpm for 5 min, and the supernatant was discarded. After carefully washing the sediment with 5 mL of water containing 3 drops of ethanol (Fisher Scientific, Loughborough, UK) and centrifuging it, 5 mL of glycerin–water (50%) was added, and the sediment was mounted in gelatin-glycerinate (VWR chemicals, Fontenay-sous-Bois, France). The frequency of the appearance of the different pollen types was divided into the following classes: dominant pollen (>45% of the pollen spectrum); accompanying pollen (15–45%); important pollen (3–15%); minor pollen (1–3%); and other pollen (≤1%).

### 2.3. Honey Processing and Storage

The raw honey was manually homogenized inside a laminar flow chamber (BioSafety Cabinet Telstar Bio II Advance, Terrassa, Spain). For each processing condition (HPP at 600 MPa for 5, 15, and 30 min and TP at 78 °C for 6 min and at 55 °C for 15 min), 145 g of honey was placed into low-permeability polyamide-polyethylene bags (PA/PE-90, Albipack-Packaging Solutions, Portugal) and thermosealed using a vacuum packager (Packman, Albipack, Águeda, Portugal) with care to avoid, as much as possible, air bubbles inside the bags. In this way, this honey was used for HPP and TP (Table 1). In total, three independent experiments were carried out with honey from the same batch.

After the processing, the processed and unprocessed honey was distributed inside a laminar flow chamber into UV-light-sterilized 50 mL transparent glass pots. In this way, the processed and unprocessed honey samples were stored (for 6, 12, and 24 months) and used in our analyses; for the analyses performed at day 0, the honey was placed into plastic tubes and stored at −80 °C and analysed for up to a max. of 2 months (except the microbial analyses, which were performed immediately after the processing).

#### 2.3.1. High Pressure Processing at Room Temperature

The HPP unit used in this experiment was a piece of industrial equipment with volume of 55 L (Hiperbaric 55 L, Hiperbaric, Burgos, Spain). The equipment consists of a 55 L volume vessel with a 200 mm diameter and an automatic loading/unloading system. The pressure was generated by pumping water into the pressure chamber. Honey samples were processed at a pressure of 600 MPa for 5, 15, and 30 min. The input water temperature was 20 ± 2 °C, the compression rate was about 250 MPa/min, and the decompression time was less than 5 s. Immediately after processing, the honey samples were placed in ice-cooled water (4 °C) to cool down.

#### 2.3.2. Conventional Thermal Processing

The thermal processing (TP) of honey samples was carried out at ambient pressure (0.1 Mpa) with a temperature of 55 °C (mild thermal processing) or 78 °C (pasteurization conditions) for 15 and 6 min, respectively, in a thermostatic water bath (FA 90, FALC Instruments, Treviglio Italy). The honey samples were fully immersed in the water bath, along with an immersion thermostat (Yellow Line, Ika, Staufen, Germany), and then immediately placed in ice-cooled water (4 °C) to cool down.

### 2.4. Microbial Analysis of Honey

For the determination of the total aerobic mesophilic microorganisms, yeasts and moulds, and total coliform microorganisms, 10 g of the honey sample was homogenized with 90 mL of sterile Ringer’s solution (Oxoid Ringers Solution Tablets, Thermo Fisher Scientific, Hampton, VA, USA), using a lab blender (Stomacher 400 Circulator, Seward, London, UK), for 2 min. Further, decimal serial dilutions were prepared from this homogenate using the same sterile diluents and plated with the appropriate media, according to the procedure that we followed. The effect of processing, on day 0, on these microbials was determined immediately after the HPP and TP.

#### 2.4.1. Yeasts and Moulds

Yeasts and moulds were enumerated on sterile selective agar (RBC agar, Difco, Detroit, MI, USA) (ISO 21527-2:2008) [30]. Briefly, 1 mL from each dilution (10^−1^, 10^−2^, 10^−3^) of the honey sample solutions was inoculated on the surface of RBC agar plates and incubated at 25 ± 1 °C for 5 days. Plates containing 1–150 colonies were selected for counting, and the results were expressed as log colony-forming units per g of honey (log CFU/g). Each sample was prepared in triplicate and with five repetitions of each decimal dilution (the total volume of sample plated per dilution was 1 mL).

#### 2.4.2. Total Aerobic Mesophiles

The enumeration of the total aerobic mesophilic bacteria was carried out on sterile agar (PCA agar, Panreac AppliChem, Barcelona, Spain). Briefly, 1 mL from each dilution (10^−1^, 10^−2^, 10^−3^) of the honey sample solutions was incorporated with PCA media. The total colonies formed were counted after incubation at 30 ± 1 °C for 48 h (ISO 4833:2013) [31]. Plates containing 15–150 colonies were selected for counting, and the results were expressed as log colony-forming units per g of honey (log CFU/g). Each sample was prepared in triplicate and with triplication of the decimal dilutions (the total volume of sample plated per dilution was 1 mL).

#### 2.4.3. Total Coliforms and *Escherichia coli*

The total coliform count was determined on selective agar (CCA, VWR International, Leuven, Belgium). Briefly, 1 mL from each sample dilution (10^−1^, 10^−2^, 10^−3^) was incorporated with the CCA media and incubated at 35 ± 1 °C for 24–48 h. Plates containing 15–150 CFU were selected for counting, and the results were expressed as log colony-forming units per g of honey (log CFU/mL). Each sample was prepared in triplicate and with triplication of the decimal dilutions (the total volume of sample plated per dilution was 1 mL).

### 2.5. Physicochemical Analysis of Honey

The physicochemical qualities of the honey were analysed using the Official Methods of Analysis of the Association of Official Analytical Chemists (AOAC, 1990) and the Harmonized Methods of the International Honey Commission (IHC) [32]. All the tests were performed in triplicate.

The moisture content (AOAC Official Method 969.38) of honey was determined using a refractometer (Atago PAL-BX/RI pocket refractometer, Tokyo, Japan). The refractive indexes of the honey samples were measured at a temperature of 25 °C; the readings were further corrected to a temperature of 20 °C using a factor of 0.00023 per °C and the results were expressed as percentages.

The pH of honey was determined from a solution of 10 g of honey in 75 mL of CO_2_-free distilled water using a pH meter (Basic20, Crison, Barcelona, Spain).

Its sugar content was determined using a refractometer (Atago PAL-BX/RI pocket refractometer, Tokyo, Japan) with a direct reading display, and the results were expressed as Brix.

The electrical conductivity of the honey solution at 20% (*w*/*v*) was measured at 20 °C using a conductivity meter (Type 522, Crison Instrument, S.A., Barcelona, Spain) equipped with a conductivity probe. The honey sample solution was prepared using distilled water. Results were expressed as mS/cm.

The water activity (aw) was measured at 25 °C using a hygrometer (Novasina LabSwift-aw, Lachen, Switzerland), and the results were expressed to two decimal places.

#### 2.5.1. Hydroxymethylfurfural Content (HMF)

The determination of HMF content was based on the spectrophotometric method of the IHC [32]. Briefly, the honey (5 g) was dissolved in approximately 25 mL of distilled water and quantitatively transferred into a 50 mL volumetric flask. Then, 0.5 mL of Carrez solution I (15 g of potassium hexacyanoferrate (II) trihydrate K_4_Fe(CN)_6_·3H_2_O diluted up to 100 mL with distilled water) and 0.5 mL of Carrez solution II (30 g zinc acetate dihydrate Zn(CH_3_COO)_2_·2H_2_O diluted up to 100 mL with distilled water) (Sigma-Aldrich, Saint-Quentin Fallavier, France) were added to the honey solution, which was then made up to its final volume (50 mL) with distilled water. The solution was filtered through filter paper and the first 10 mL of the filtrate was rejected. Then, 5 mL aliquots of the filtrate were put in two test tubes: (i) to one test tube we added 5 mL of distilled water (sample solution) and (ii) to the second tube we added 5 mL of 0.2% sodium hydrogen sulphite (Sigma-Aldrich, Saint-Quentin Fallavier, France) solution (0.20 g of NaHSO_3_ diluted up to 100 mL with distilled water). The absorbance of the sample solution against the reference solution at 284 nm and 336 nm was determined, in quartz cuvettes, using a spectrophotometer (3100-Spectrophotometer, VWR International, Radnor, PA, USA). The content of HMF was calculated by the application of the following formula:HMF (mg/kg) = [(A_284_ − A_336_) ∗ F ∗ 5]/P(1)
where A_284_ and A_336_ are the absorbance readings, F is a constant value of 149.7, and P is the weight of the sample in grams.

#### 2.5.2. Diastase Activity

The diastase activity in honey was measured using an Amylazyme assay according to Megazyme’s instructions (Megazyme International Ireland Ltd., Bray, Ireland). Briefly, 2 g of honey was dissolved in 50 mL of 100 mM sodium maleate (Sigma-Aldrich, Saint-Quentin Fallavier, France) buffer (pH 5.6), and 1 mL of the diluted honey solution was transferred into test tubes. The tubes containing honey solution were pre-incubated at 40 °C in a water bath. After 5 min of pre-incubation, an Amylazyme tablet was added into each test tube, without stirring, and they stood, for incubation at 40 °C, in the water bath for exactly 10 min. Then, 10 mL of Trizma base (Sigma-Aldrich, Saint-Quentin Fallavier, France) solution (2%, *w*/*v*) was added and vigorously stirred in. After approximately 5 min, the test tubes were stirred again and then filtrated through filter paper (Whatman No. 1). The absorbance of the sample solution was measured at 590 nm against a reaction blank. The reaction blank was prepared by adding the Amylazyme tablet into sodium maleate buffer and proceeding as with the sample solutions. The diastase activity of the honey sample was determined by use of the associated regression equation, which is as follows:Diastase activity (DN) = 20 ∗ A_590_
(2)
where A_590_ is the absorbance at 590 nm. The results were expressed as a diastase number (DN).

### 2.6. Total Phenolic Content (TPC)

The TPC of honey samples was analysed using a microplate method based on the 96-well microplate Folin–Ciocalteu method, according to the methodology described by Bobo-Garcia et al. (2015) [33]. Folin–Ciocalteu reagent (Sigma-Aldrich, Saint-Quentin Fallavier, France) was diluted (1:4) and 100 µL was mixed with 20 µL of a honey sample (0.1 g/mL) in a flat-bottom 96-well microplate. After 4 min, 75 µL of sodium carbonate (Sigma-Aldrich, Saint-Quentin Fallavier, France) solution (100 g/L) was added and incubated for 2 h, in the dark, at 20 °C. Its absorbance was recorded at 750 nm using a microplate spectrophotometer (Multiskan GO, Thermo Fisher Scientific Inc., Hampton, VA, USA). Gallic acid (Sigma-Aldrich, Saint-Quentin Fallavier, France) was used as the standard (0–200 mg/L) for calibration. The blank solution consisted of methanol–water (70:30, *v*/*v*) instead of honey or standard and was subtracted from the absorbance of the reaction with the honey sample. The results were expressed as mg gallic acid equivalents (GAE)/100 g of honey.

### 2.7. Determination of Honey Antioxidant Activity

#### 2.7.1. DPPH Radical Scavenging Activity Assay

The 1,1-diphenyl-2-picrylhydrazyl (DPPH) radical scavenging activity of honey samples was evaluated using a microplate antioxidant activity methodology based on a 96-well plate assay described elsewhere [33]. Briefly, 20 µL of the honey sample (0.1 g/mL) was added to 180 µL of DPPH (Sigma-Aldrich, Saint-Quentin Fallavier, France) solution (150 µmol/L) in methanol–water (80:20, *v*/*v*) and shaken for 60 s in a 96-well microplate. The absorbance was measured at 515 nm, after 40 min of incubation at 20 °C in the dark, using a microplate spectrophotometer (Multiskan GO, Thermo Fisher Scientific Inc., Hampton, VA, USA). The DPPH scavenging activity was calculated using the following equation:(3)$$DPPH\;\%\;inhibition=\left\lfloor 1-\left(\frac{A_{sample-}A_{blank}}{A_{control}-A_{blank}}\right)\right\rfloor \times 100$$
where A_sample_ is the absorbance at 515 nm of 20 µL of a honey sample in 180 µL of DPPH solution after 40 min of incubation; A_blank_ is the absorbance at 515 nm of 20 µL of water and 180 µL of methanol–water (80:20, *v*/*v*) after 40 min; and A_control_ is the absorbance at 515 nm of 20 µL water and 180 µL of DPPH solution after 40 min.

#### 2.7.2. Ferric Reducing Antioxidant Power Assay (FRAP)

The reducing capacity of honey samples was determined based on the method of Benzie and Strain, adjusted to a microplate FRAP assay according to the modification of Bolanos de la Torre et al. (2015) [34]. The FRAP working solution consisted of a mixture of 10 parts acetate buffer (300 mM, pH 3.6); 1 part 10 mM 2,4,6-tripyridyl-s-triazine (Trizma, Sigma-Aldrich, Saint-Quentin Fallavier, France) dissolved with 40 mM HCl (Sigma-Aldrich, Saint-Quentin Fallavier, France); and 1 part 20 mM iron (III) chloride hexahydrate (FeCl_3_·6H_2_O, Panreac, Barcelona, Spain) dissolved in distilled water, and was then warmed up to 37 °C 10 min before use. Then, 280 µL of this FRAP working solution was mixed with 20 µL of a honey sample solution (0.1 g/mL) and then incubated at 37 °C for 30 min in the dark and read at 593 nm using a microplate reader (Multiskan GO, Thermo Fisher Scientific Inc., Hampton, VA, USA). Trolox was used as the standard (0–1000 µM/L) to obtain the calibration curve. The blank solution consisted of 20 µL of distilled water and 280 µL of the FRAP solution, and the absorbance of the blank was subtracted from the absorbance of the reaction with the honey sample. Honey antioxidant activity was expressed as 10^−6^ M Fe (II) divided by the 10% solution of honey (µM Fe^2+^/10% honey).

### 2.8. Statistical Analyses

The data of the obtained results for each honey sample (unprocessed honey, HPP 600/5, HPP 600/15, HPP 600/30, TP 55/15, and TP 78/6), from day 0 and each storage period, were expressed as a mean value, with standard deviation (SD), of three independent experiments with triplicated measurement.

A two-way ANOVA analysis with Turkey’s post hoc test was used to determine the differences between thermally/HPP-treated and untreated honey samples. The relationship between the measured antioxidant properties was calculated using Sperman’s ran correlation coefficient. Data with *p*-values smaller than 0.05 were considered statistically significant. All statistical analyses were performed using GraphPad Prism version 9.2.0 (GraphPad Software Inc., La Jolla, CA, USA).

## 3. Results

### 3.1. Characterization of the Honey’s Botanical Origin

The determination of the honey’s botanical origin was based on the relative frequencies of the appearance of pollen types belonging to nectariferous species. The average pollen content of the studied honey is depicted in Figure 1, along with the scientific names of the species seen and their pollen frequency.

The highest pollen content in the honey was found to be *Trifolium pratense* L. (red clover), with a pollen representation of 42% of the total pollen grains quantified, making it the dominant pollen. However, its % frequency was below the minimum percentage of pollen required (>45%) for the honey’s characterization as a monofloral clover honey. In addition, pollens from Portuguese broom (*Cytisus scoparius*), viper’s bugloss (*Echium vulgare*), oak (*Quercus faginea*), chestnut (*Castanea sativa*), lavender (*Lavandula stoechas*), and cherry hill (*Prunus serrulate*) were found in the honey in the range of 3–15%, contributing important amounts of pollen to the honey. Given that clovers are the predominant flora in pasture honey [35], the honey used in this study was classified as multifloral pasture honey.

### 3.2. Characterization of the Honey’s Physicochemical Properties

Table 2 shows the results obtained from the physicochemical analysis of the raw honey sample, including its moisture, sugar content, water activity, electrical conductivity, and pH.

The moisture content of the honey sample suggests that the honey achieved a mature state within the hive. Also, based on its °Brix value, the honey showed a good maturity degree, as it was above 60 °Brix. The pH value of the honey was acidic and within the standard limits (3.4–6.1) of the international standard [2]. The electrical conductivity value determined in the honey sample was within the standard limit for blossom-type honey (less than 0.8 mS/cm). All the values of the honey’s measured physicochemical properties were within the legal limits established by the European Directive 101/2001 [1].

### 3.3. Effect of HPP and TP on the Microbial Load of Honey

The results in terms of the microbial load in unprocessed honey and HPP-treated and TP-treated honey, before and after storage for a period of 6, 12, and 24 months, are shown in Figure 2. The microbial load of the unprocessed honey samples was 4.10 ± 0.08 and 3.80 ± 0.09 log CFU/g for yeasts/moulds and aerobic mesophiles, respectively. Coliforms and *E. coli* were not detected. All HPP-treated honey samples showed no significant differences in their microbial load compared to untreated honey. Similarly, the microbial load of mildly thermally processed honey samples remained unchanged, with an average value of 3.92 ± 0.03 and 3.85 ± 0.01 log CFU/g for yeasts/moulds and aerobic mesophiles, respectively. However, the pasteurization (78 °C/6 min) of the honey sample resulted in a significant decrease in the loads of yeasts/moulds and aerobic mesophiles to values of 1.75 ± 0.05 and 2.09 ± 0.12 log CFU/g, respectively.

Over the storage period, the unprocessed honey sample showed a decrease in microbial counts for all microorganisms analysed, where the yeasts/moulds and aerobic mesophiles counts decreased to values of 2.46 ± 0.04 and 2.78 ± 0.02 log CFU/g, respectively, after 24 months of storage. The storage of pasteurized honey for a period of 6 months resulted in a total inactivation (<1 log CFU/g) of its yeasts/moulds and aerobic mesophiles. No differences in microbial load between the mildly thermally treated honey and untreated honey were found after 6 months of storage. On the other hand, all HPP-treated honey samples showed a significant decrease (*p* < 0.05) in their yeasts/moulds load compared to unprocessed honey, after 6 months of storage (Figure 2A).

Interestingly, all the HPP samples achieved a commercially acceptable microbial level (below 2.7 log CFU) after 12 months of storage, with values of 2.35 ± 0.06, 2.38 ± 0.06, and 2.46 ± 0.07 log CFU/g for HPP 600/5, HPP 600/15, and HPP 600/30, respectively, while the average values for the unprocessed honey and TP 55/15 honey were 3.42 ± 0.08 and 3.07 ± 0.05 log CFU/g, respectively. Moreover, all HPP-treated honey samples stored for a period of 24 months showed a total inhibition (<1 log CFU/g) of yeasts/moulds and aerobic mesophiles, with their values equivalent to values of the thermally pasteurized honey samples. The TP 55/15 honey samples showed a significantly lower microbial reduction at 24 months of storage for yeasts/moulds and aerobic mesophiles, with values 1.67 ± 0.07 and 1.48 ± 0.26 log CFU/g, respectively. HPP accelerated the decrease in yeasts/moulds and aerobic mesophiles in honey during storage compared to unprocessed honey and honey thermally treated under mild conditions (55 °C/15 min).

### 3.4. Effect of HPP and TP on the HMF Content and Diastase Activity in Honey

The effect of HPP and TP on selected qualitative honey criteria, namely its HMF content and diastase activity, is presented in Figure 3. The initial content of HMF in the honey sample was 6.69 ± 0.35 mg/kg and its diastase activity, expressed as a DN value, was 18.23 ± 0.25. The pasteurization of honey samples significantly increased their HMF content and the subsequent storage of pasteurized as well as mildly thermally treated honey samples resulted in a further gradually increased HMF content in the honey. The permissible level of HMF (40 mg/kg) was exceeded in the pasteurized honey after 12 months of storage as well as in the mildly thermally treated honey after 24 months of storage, reaching average HMF values of 46.37 ± 1.45 and 57.16 ± 3.99 mg/kg, respectively. The HMF content of the HPP-treated honey neither changed immediately after HPP nor changed over the entire storage period, compared to untreated honey (Figure 3A).

In the case of diastase activity, both the mild thermal treatment and pasteurization caused a significant reduction in enzymatic activity by 6.5 and 15.5%, respectively, and the storage of these thermally processed samples further decreased their DN value. The DN value of the pasteurized honey sample was found to be under the permissible level (DN = 8) after 12 months of storage. Although the storage of mildly thermally treated honey samples gradually significantly decreased their DN value, their average values after 12 and 24 months of storage were above the permissible level for diastase activity. The level of the diastase activity of HPP-treated honeys neither changed immediately after HPP nor changed over their entire storage period, compared to untreated honey (Figure 3B).

The HMF concentration in untreated honey samples gradually changed with increasing storage time; however, its level was found to be under the permissible level during the storage period.

### 3.5. Effect of HPP and TP on the Honey’s Antioxidant Properties

Changes in the TPC of honey and the honey’s antioxidant properties, determined by DPPH and FRAP assay, were evaluated immediately after HPP or TP and after the storage of processed and untreated honey samples for periods of 6, 12, and 24 months (Figure 4). Both HPP and TP did not affect the TPC and the average values of TPC did not differ after storage for 6 months. Statistically significant changes were observed after 12 months of storage, where the TPC was higher in the HPP-treated sample (HPP 600/5; *p* < 0.05) but significantly lower in the pasteurised honey sample (*p* < 0.01) compared to the untreated sample. The average value of the TPC in the HPP-treated sample (HPP 600/5; *p* < 0.01) was further increased by 23% compared to the untreated sample after 24 months of storage.

The effect of HPP and TP on the honey’s antioxidant activity is shown in Figure 4B,C. Both processing methods did not significantly affect the antioxidant activity of the honey sample, except for the mild thermal treatment, for which a significantly higher FRAP value (*p* < 0.05) was documented. However, a longer storage of the processed honey samples resulted in a significant change in their antioxidant activity. Thermal-treated honey samples, in particular pasteurised samples, showed augmented antioxidant activity, evaluated by DPPH assay, after 12 and 24 months of storage. The DPPH radical-scavenging activity of the pasteurised honey samples was elevated by 39 and 64% compared to the untreated samples after 12 and 24 months of storage, respectively. On the contrary, thermally treated stored honey samples had significantly lowered FRAP values, by 21 and 33%, compared to the untreated honey samples after 12 and 24 months of storage, respectively.

The DPPH activity, as well as the ferric reducing ability of the HPP-treated honey samples, after their storage for 6, 12, and 24 months, remained unchanged when compared to untreated honey samples, except for the processed samples HPP 600/15 and HPP 600/30, where a significant increase in DPPH activity, by 15 and 24%, respectively, was observed after 24 months of storage.

The statistical analysis did not reveal any correlation (12 months: r_s_ = −0.821, *p* = 0.058; 24 months: r_s_ = −0.771, *p* = 0.103) between the TPC content and DPPH activity of all processed honey samples after 12 and 24 months of storage. On the other hand, a statistically significant correlation (12 months: r_s_ = 0.342, *p* = 0.017; 24 months: r_s_ = 0.841, *p* = 0.044) was found between the TPC content and FRAP activity of all processed honey samples after 12 and 24 months of storage.

## 4. Discussion

The thermal treatment of honey is widely used in industrial honey as a type of processing that avoids honey’s fermentation and delays its crystallization. Furthermore, it is a traditional method used to liquify crystallized honey. Due to its frequent usage and potential detrimental effect on honey’s quality and its biological properties, overheating or the prolonged thermal processing of honey needs to be monitored by determining the values of its DN and HMF. Therefore, alternative methods of honey pasteurization and preservation, such as ultrasound and HPP technology, are being tested to maintain honey’s initial freshness and biological functionality and to improve its shelf life. In addition, HPP might be preferably applied to honey samples with a naturally higher moisture content, thus reducing their risk of fermentation.

In this study, we employed HPP technology, which processed honey samples at a pressure of 600 MPa for 5, 15, and 30 min. A conventional thermal treatment of honey samples was carried out at 55 and 78 °C for 15 and 6 min, respectively. We found that the pasteurization of honey at 78 °C for 6 min significantly decreased its initial yeasts/moulds and mesophilic aerobes loads by 58 and 45%, respectively. The HPP of honey samples was not as effective in its inactivation of microorganisms as thermal pasteurization, and there were no changes in these samples’ microbial load after processing compared to untreated honey. However, a significant decrease in the microbial loads in HPP-treated samples compared to untreated samples was observed with the increasing length of honey storage. These results indicate that thermal pasteurization has an immediate effect on the viability of microorganisms in honey, while HPP did not exhibit such a lethal effect immediately on the microorganisms in honey. The efficacy of HPP on microbial inactivation depends on the food’s composition and properties. High contents of sugars, fats, or salt in food have baroprotective effects on microbial cells [36]. Recently, Fauzi et al. (2017) found that *Saccharomyces cerevisiae* cells in honey exhibited a gradual increase in their resistance to HPP with an increasing sugar concentration [22]. In addition, the pH of the food and its water activity, as well as the type of microorganisms to be inactivated, including their taxonomic unit and strain, also significantly affect the efficacy of HPP [37]. Although the exact mechanism of HPP’s antimicrobial action has not yet been fully elucidated, HPP can cause the rupture of cell walls, membrane damage to microbial cells, and damage to membrane proteins, enzymes, and ribosomes [38]. Besides a lethal inactivation of microorganisms, HPP can induce a wide range of injuries in the cell population [39]. Thus, the HPP-mediated sublethal inactivation of cells generates injured cells which may recover or be further injured depending on the severity of their environmental conditions. Our results showed that the yeast/mould and aerobic mesophile loads in HPP-treated samples significantly decreased after 6 and 12 months, respectively, suggesting the vital role of honey’s compositions (antimicrobial compounds) and physiochemical properties (e.g., low pH, low aw) in the killing of HPP-mediated injured microbial cells. Concerning pH values, it has been reported that the microbial inactivation by HPP was increased when the pH value was low [40]. In the case of honey, its pH is low, with values ranging between 3.2 and 4.5 [41]. Besides its low pH and water activity, raw honey contains antibacterial substances such as bee defensin-1, an antibacterial peptide [42,43] and glucose oxidase enzyme responsible for generating H_2_O_2_ in diluted honey [44].

Over the last few decades, numerous studies have investigated the effect of HPP on protein denaturation and aggregation [45]. Contrary to thermal processing, where covalent and non-covalent chemical bonds are disrupted, HPP can disrupt only weak chemical bonds including hydrogen and hydrophobic bonds [46]. Thus, the primary structure of proteins is not affected by HPP; however, the protein’s secondary structure can be reversibly/irreversibly modified depending on the HPP conditions and the proteins’ properties [45]. Similarly, the primary structures of low-molecular-weight molecules such as amino acids, vitamins, and pigments are not affected by HPP.

In this study, the enzymatic activity of bee-derived diastase, expressed as the DN value, was evaluated in untreated and treated honey samples. As expected, the thermal treatment of honey at both temperatures (55 and 78 °C) resulted in a significant decrease in diastase’s enzymatic activity after 6, 12, and 24 months of storage. At the higher temperature of 78 °C, the loss of the enzymatic activity of diastase was observed immediately after thermal processing. Furthermore, the DN values in honey samples treated at 78 °C were lowered below the permissible level after 12 and 24 months of storage. On the other hand, no significant changes in the diastase’s enzymatic activity were observed in HPP-treated honey samples immediately after HPP and after storage for 6, 12, and 24 months, when compared to untreated honey samples. These results agree with previous research, where the application of HPP with comparable conditions did not change the enzymatic activity of stingless bee honey and manuka honey’s diastase immediately after processing [25,27]. In the case of untreated honey, long-term storage caused a slight decrease in its DN values, but the values remained within the acceptable quality level. It is well documented that temperature and time are the main drivers of the decrease in diastase’s enzymatic activity in honey; however, the specific phytochemicals of certain types of honey can accelerate the loss of diastase’s enzymatic activity [47].

Since HMF is not naturally present in freshly collected honey and is formed upon thermal processing or prolonged storage, its concentration has been evaluated only in a few studies characterizing the effect of HPP on honey [24,25]. Overall, HPP, at ambient temperatures, does not increase the concentration of HMF in honey, as was shown here and elsewhere [24,25]. Moreover, the content of HMF in HPP-treated honey samples did not change compared to untreated samples upon long storage (12–24 months). In the case of untreated honey samples, a significant increase in their HMF content was observed after 12 and 24 months of storage; however, their HMF values were within the permissible range. It has been shown recently that a higher water content, lower pH, and higher Ca^2+^ and Mg^2+^ contents are the main factors responsible for increased HMF formation in honey during its storage [48]. Furthermore, the HMF content greatly varied in different types of honey after its thermal treatment at the same temperature and duration [49]. Antioxidant activity is one of the most often studied of honey’s biological properties, and several different methodological approaches are used to determine its antioxidant activity, such as radical scavenging activity assays using 1,1-diphenyl-2-picrylhydrazyl (DPPH), oxygen radical absorbance capacity (ORAC) assays, and ferric reducing antioxidant power (FRAP) assays [50]. The thermal treatment of honey with diverse temperature and duration conditions and different types of honey has led to conflicting results [50]. In most cases, heating reduced the antioxidant capacity of honey and lowered its TPC content. On the other hand, some studies showed an increase in its antioxidant activity, most likely resulting from the production of Maillard reaction products with antioxidant activity during thermal processing [51,52]. In the present study, the antioxidant activity of thermally pasteurized honey samples remained unchanged or slightly increased at 55 °C (FRAP activity). However, the storage of untreated and treated honey samples resulted in the significantly increased DPPH activity of the honey samples treated at a temperature of 78 °C, over the storage time. On the contrary, the antioxidant capacity of stored honey samples thermally treated at 78 °C, determined by FRAP assay, significantly decreased with the increasing length of storage. This observed discrepancy can be explained by the presence of different Maillard reaction products such as intermediates, volatile flavour compounds, and melanoidins which, possess different mechanisms of antioxidant action [53]. An important factor that determines the increased or decreased antioxidative capacity of stored thermally treated honey is the type of honey and its composition. Recently, Flanjak et al. (2022) characterized the antioxidant activity of thermally treated (45 °C/48 h and 65 °C/6 h) and stored (up to 24 months) sage (*Salvia officinalis* L.) honey [54]. The processing and storage of sage honey increased its TPC and FRAP values but decreased its DPPH values over its storage time. The formation and accumulation of Maillard reaction products, such as Amadori compounds and α-dicarbonyl compounds, in honey during its storage gradually increased, but these increasing rates differed significantly [55]. The exact composition of the Maillard reaction products in honey may differ from honey to honey and may be affected by processing and storage conditions. Therefore, the antioxidant capacities of thermally or non-thermally processed and/or stored honey may be significantly different.

## 5. Conclusions

Firstly, the qualitative criteria of unprocessed raw honey after its long-term storage (24 months) did change, but the measured values were within the legislatively determined permissible level. Furthermore, the antioxidant activity of unprocessed raw honey remains stable within a storage period of 24 months.

Although pasteurization (78 °C/6 min) of the honey sample resulted in the immediate inactivation of the microorganisms in honey, it showed detrimental effects on the qualitative criteria and antioxidative properties of the stored processed honey samples. Both qualitative criteria, HMF and DN, did not meet the requirements of the European Honey Directive after the 12- and 24-month storage of pasteurised honey samples. On the other hand, HPP was likely to cause only sublethal changes/damages to the honey’s microorganisms, but the subsequent storage of HPP-treated samples resulted in the lethal inactivation of honey’s microbial content due to the honey’s composition. The contrary to pasteurization, the storage of HPP-treated honey samples neither changed their qualitative criteria nor significantly decreased the honey’s antioxidant activity. Interestingly, HPP showed a tendency to slightly increase the TPC of prolongedly stored honey (24 months). Therefore, HPP was found to be a more suitable and safe preservation method for honey, with a minimal loss of the honey’s antioxidant activity and a tendency to decrease the honey’s TPC by less than traditional industrial honey pasteurization.

## Figures and Tables

**Figure 1 foods-13-00989-f001:**
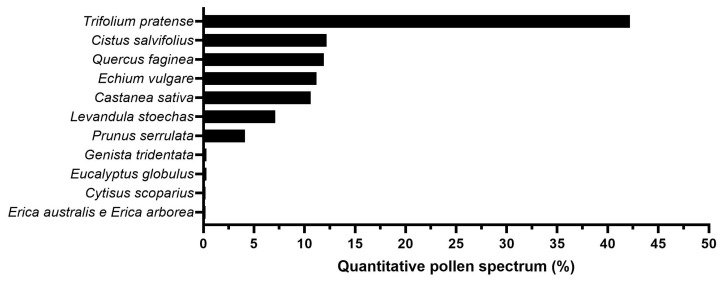
Botanical origin of honey: dominant pollen (>45% of the pollen spectrum); accompanying pollen (15–45%); important pollen (3–15%); minor pollen (1–3%); and other pollen (≤1%).

**Figure 2 foods-13-00989-f002:**
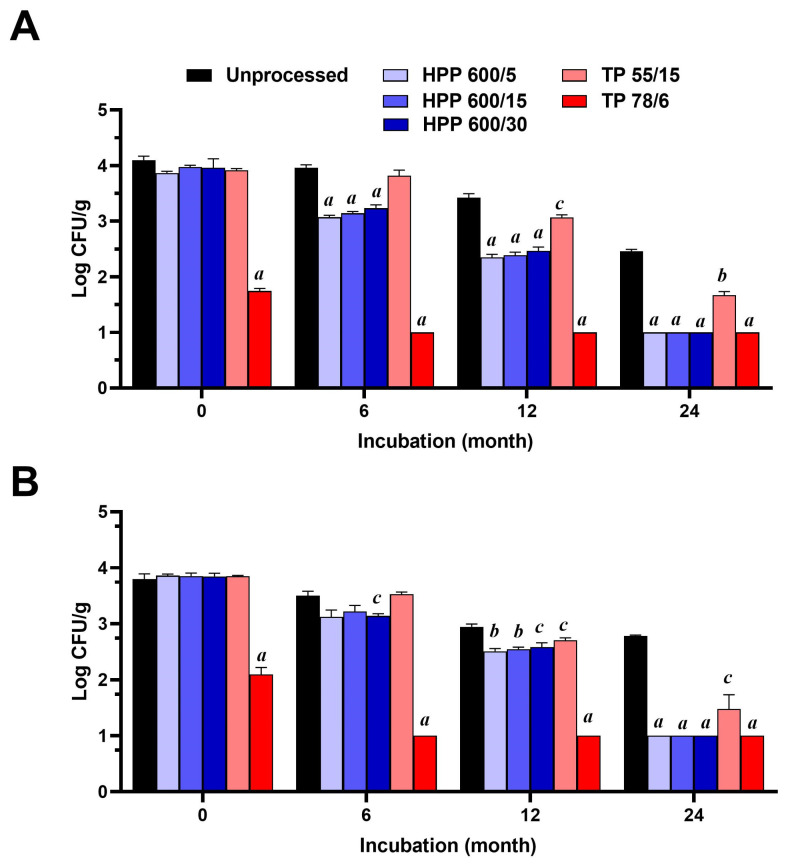
Changes in the microbial load of samples of untreated honey and honey that underwent high-pressure processing (HPP) and thermal processing (TP) (*n* = 3) under different conditions, expressed as colony forming unit per g of honey (CFU/g), over a storage period of 24 months. (**A**) Load of yeasts/moulds. (**B**) Load of aerobic mesophiles. The data are expressed as the mean of CFU values. a, *p* < 0.001; b, *p* < 0.01; c, *p* < 0.05.

**Figure 3 foods-13-00989-f003:**
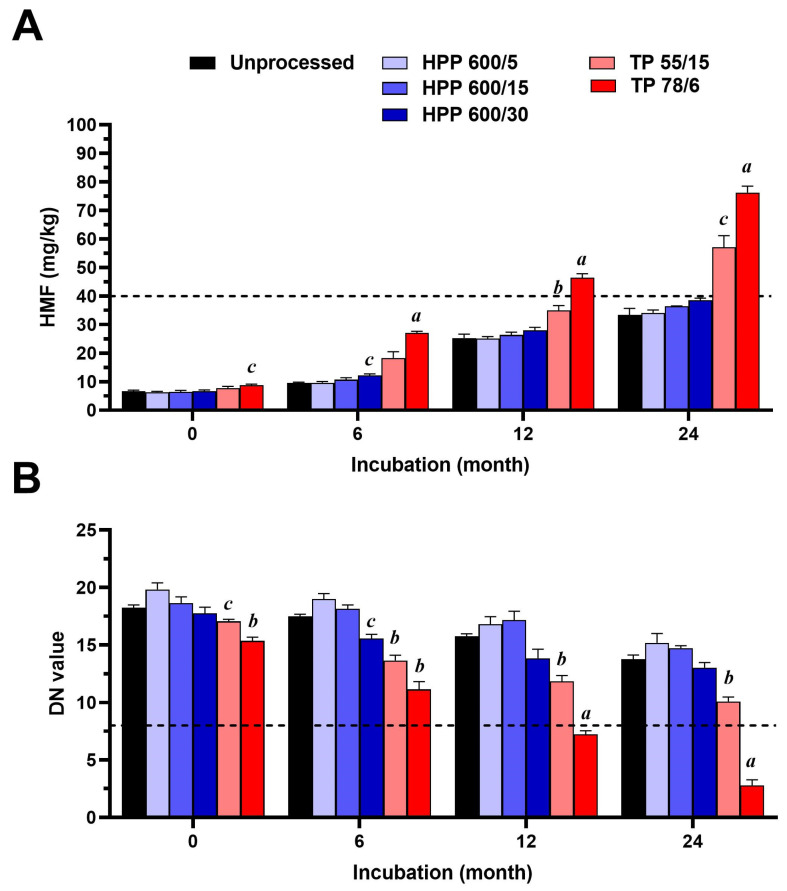
Changes in hydroxymethylfurfural (HMF) content and diastase activity (DN) in samples of untreated honey, and honey that underwent high pressure processing (HPP) and thermal processing (TP) (*n* = 3) under different conditions, over the storage period of 24 months. (**A**) HMF content. (**B**) DN. The data are expressed as the mean of HMF or DN values. The dotted line represents the permissible legislative level for these qualitative criteria. a, *p* < 0.001; b, *p* < 0.01; c, *p* < 0.05.

**Figure 4 foods-13-00989-f004:**
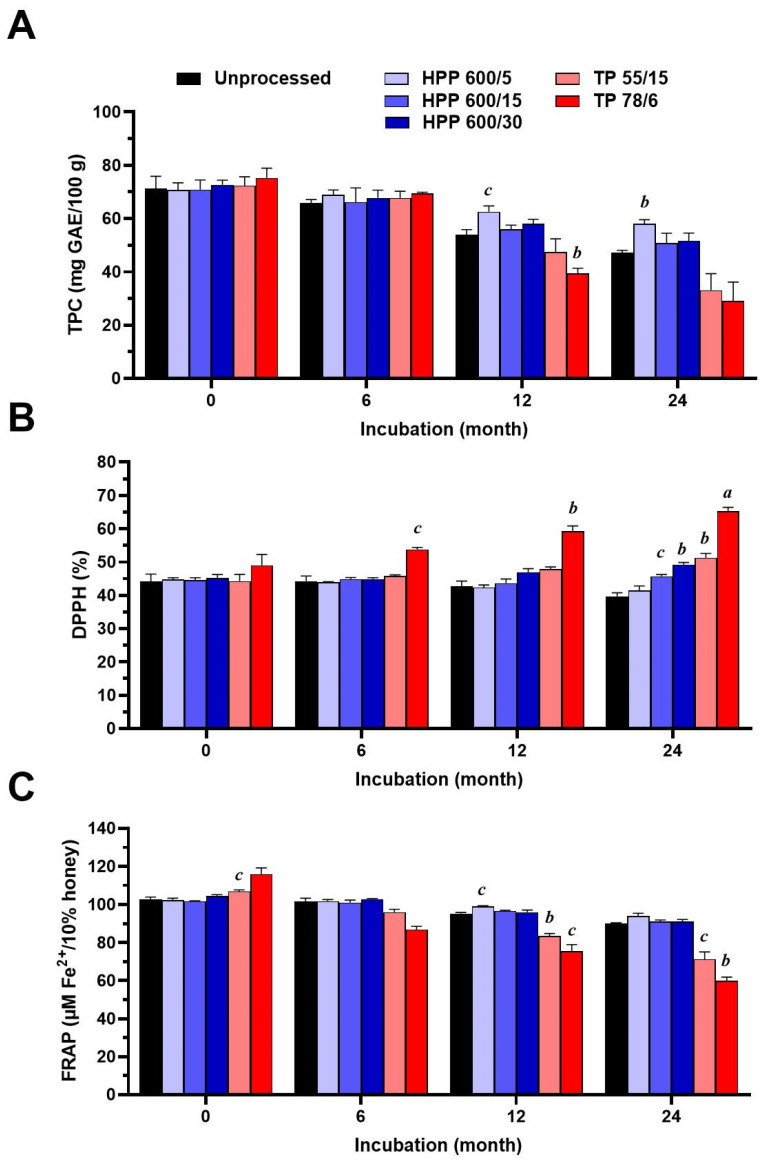
Changes in the total polyphenol content (TPC) and antioxidant activity of samples of untreated honey, and honey that underwent high-pressure processing (HPP) and thermal processing (TP) (*n* = 3) under different conditions, over the storage period of 24 months. (**A**) TPC. (**B**) 2,2-Diphenyl-1-picrylhydrazyl (DPPH) antioxidant assay. (**C**) Ferric reducing antioxidant power (FRAP) assay. The data are expressed as the mean of their TPC, DPPH, and FRAP values. a, *p* < 0.001; b, *p* < 0.01; c, *p* < 0.05.

**Table 1 foods-13-00989-t001:** Experimental conditions for honey HPP and TP. All honey samples were from the same honey batch. The honey samples were analysed immediately after processing (day 0), and after 6, 12, and 24 months of storage at a controlled temperature of 20 °C.

	Honey Processing Conditions				
Processing Parameters	HPP 600/5	HPP 600/15	HPP 600/30	TP 55/15	TP 78/6
Pressure (MPa)	600	600	600	0.1	0.1
Time (min)	5	15	30	15	6
Temperature (°C)	22	22	22	55	78

HPP, high pressure processing; TP, thermal processing.

**Table 2 foods-13-00989-t002:** Physicochemical properties of tested honey (n = 3).

Physicochemical Parameters	Mean ± SD
Moisture (%)	14.50 ± 0.18
Sugars (°BRIX)	85.00 ± 0.12
water activity	0.58 ± 0.00
EC (mS/cm)	0.45 ± 0.01
pH	3.97 ± 0.06

EC, electrical conductivity; SD, standard deviation.

## Data Availability

The original contributions presented in the study are included in the article, further inquiries can be directed to the corresponding author.

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
