# Peer review of "The High Pressure Preservation of Honey: A Comparative Study on Quality Changes during Storage"

_foods, 2024, doi:10.3390/foods13070989_

Round 1

Reviewer 1 Report

Comments and Suggestions for Authors

The manuscript presents a study that evaluates the impact of high-pressure processing (HPP) and thermal processing on the quality and antioxidant capability of honey during storage. The research aims to identify a preservation method that maintains honey's beneficial properties. The study includes various analyses, such as microbial load, qualitative parameters (HMF content and diastase activity), and antioxidant activity (DPPH and FRAP assays), providing a comprehensive evaluation of honey properties.

* Lack of Specific Details: The manuscript lacks specific details regarding the HPP conditions used, such as the pressure level and processing time. Providing this information would enhance the understanding of the study's methodology.

* The concept of using HPP as a honey preservation method has been previously explored in other studies. 

* what about the enzyme activity that proves the honey is natural?

Recommendations:

 - Expand on Methodology: Provide more details about the HPP conditions used, including pressure level, processing time, and any additional parameters that were controlled during the study.

 - Include Key Findings: Summarize the main findings of the study, such as the specific changes observed in microbial load, qualitative parameters, and antioxidant activity after HPP and thermal processing, along with the statistical significance of the results.

 - Highlight Novelty: If the study offers novel insights or unique contributions to the field, emphasize these aspects in the abstract to distinguish it from previous research.

 - Consider a Broader Impact Statement: Briefly mention the potential implications or applications of the study's findings, such as the potential benefits of HPP for the honey industry or consumers.

 Overall, the manuscript provides a solid foundation for a research article, but it would benefit from additional details regarding the methodology, a more comprehensive discussion of the results, and a clearer articulation of the study's novelty and broader impact.

Author Response

Thank you for your comments and suggestion. Please find the comments of author in the word document. 

Please check the manuscript with the highlighted changed made as red-colour text. Also, we improved English grammar (track changes) and corrected some statements.

Thank you for considering our manuscript.

Yours sincerely,

Leticia Estevinho

Reviewer 2 Report

Comments and Suggestions for Authors

I have read the manuscript and found it well written and interesting. The experiment is well-designed. I just have some minor suggestions or questions:

-why the authors chose FRAP for measuring the reducing power with non-typical (acidic) pH for such product nor relevant to physiological conditions of human organism, instead of e.g. CUPRAC method? Is it because of some kind of tradition for honey testing, as DPPH is usually chosen?

-line 329: dominant?

-I cannot agree with increased TPC in some samples (Conclusions) - it is rather "less decreased", so higher than other samples; please mind the enzymatic activity remained in your samples (as shown by diastase activity) that could help explaining the retention of phenolics.

Author Response

(The authors gave the same response as above.)

Reviewer 3 Report

Comments and Suggestions for Authors

The manuscript “High-Pressure Preservation of Honey: A Comparative Study on 2 Quality Changes During Storage” provides new and interesting results, so the paper need some changes according to the following recommendations:

- The paper does not provide information that high-quality raw honey from Apis mellifera does not need any processing and has a minimum shelf life of 24 months without the need for heat treatment. This is clearly seen from the results obtained in the control experiments. In the manuscript, the need for thermal processing of honey is exaggerated and it should be better described in which cases it is necessary. It is not clear from the studies presented what the effect of HPP will be on honey of lower quality and/or higher water content, such as unripe honey or such from stingless bees. The introduction should be rewritten accordingly.

- The abstract and discussion should be also rewritten accordingly.

The conclusion should outline results and areas of application.

Comments on the Quality of English Language

- The manuscript needs some proofreading in English by a native speaker or professional linguist, e.g. phrases  such as “avoid of fermentation”, “which may worsening”, “in compared to” etc. 

Author Response

(The authors gave the same response as above.)

Reviewer 4 Report

Comments and Suggestions for Authors

Major revision

1.      The authors described “The raw honey samples, 150 g in duplicate”. Usually, samples were in triple repetition. In addition, raw honey sampled would be better if the author selected from similar region where is similar flower resources because it will be difference in honey quality if the flower resources were different.

2.      The difference of primary composition of honey should be determined under the different storage conditions such as glucose.  

Minor revision

1. the incubation should be incubation.

Author Response

(The authors gave the same response as above.)

Reviewer 5 Report

Comments and Suggestions for Authors

the research topic is very interesting and the author's responses to it are satisfactory. they use the well-known fact that honey is almost non-perishable food but during the transfer from producer to the consumer it can be stored in inappropriate condition and deteriorate its quality. the introduction is very informative with enough information about all the parameters that can be affected by non-adequate storage. the authors choose premises to examine how high pressure and high temperature affect the honey quality during 24 months of storage. the materials and methods give enough information about experiments and the reader can easily repeat it. the results are presented well and conclusions are supported by them.

but there is few points for concern:

first of all, when there is the determination of parameters the authors should give LOD, LOQ linearity, and recovery of the method.

the second concern is about the explanation of the obtained results especially the increase of the HMF and decrease of diastase activity should be improved. 

Comments on the Quality of English Language

English style and grammar are satisfactory and the paper is easy to read and understand

Author Response

(The authors gave the same response as above.)

Round 2

Reviewer 1 Report

Comments and Suggestions for Authors

Thank you for your replying. 

Author Response

Dear reviewer,

The authors acknowledge your time for considering our manuscript.

Yours sincerely,

Leticia Estevinho

Reviewer 3 Report

Comments and Suggestions for Authors

The manuscript "Storage of honey under high pressure: A comparative study of quality changes during storage" has had some changes made to consider the reviewer's comments, so that the paper is now better structured and can be published after small change:

Since the comparison question with honeys that are of lower quality, higher water content and/or need processing has not yet been provided, the recommendation is to move the last sentence in the conclusion to the discussion section, for it cannot be taken as a conclusion without being proved.

Author Response

Dear reviewer,

Thank you for your time and consideration of our manuscript. We have incorporated your suggestion by relocating the final sentence of the conclusion to the discussion section (Lines 493-495). The pertinent modification has been highlighted in red for your ease of reference.

Yours sincerely,

Leticia Estevinho

Reviewer 5 Report

Comments and Suggestions for Authors

the authors made an effort to improve the quality of the paper and follow the reviewer's suggestions. and they did it adequately. the quality of the paper is improved and now it fulfills minimum requirements for publication.

Comments on the Quality of English Language

english style and grammar are fine without any major concerns

Author Response

Dear reviewer,

Thank you for your time and consideration of our manuscript.

Yours sincerely,

Leticia Estevinho